# Effects of Exposure to Indoor Fine Particulate Matter on Atopic Dermatitis in Children

**DOI:** 10.3390/ijerph182111509

**Published:** 2021-11-01

**Authors:** Young-Min Kim, Jihyun Kim, Seoung-Chul Ha, Kangmo Ahn

**Affiliations:** 1Environmental Health Center for Atopic Diseases, Samsung Medical Center, Seoul 06351, Korea; ymkim0218@gmail.com (Y.-M.K.); narimy@hanmail.net (J.K.); 2Samsung Medical Center, Department of Pediatrics, Sungkyunkwan University School of Medicine, Sungkyunkwan University, Seoul 06351, Korea; 3SENKO Co., Ltd., Osan 18111, Korea; scha@senko.co.kr

**Keywords:** fine particulate matters, atopic dermatitis, children, indoor, season

## Abstract

This study aimed to investigate the short-term effect of exposure to indoor fine particulate matter (PM_2.5_) on atopic dermatitis (AD) symptoms in children. Sixty-four children (40 boys and 24 girls) with moderate-to-severe AD, aged under 18 years were enrolled in the study. They were followed up from February 2019 through November 2020. Exposure to indoor PM_2.5_ in each household of the enrolled children and their AD symptoms were measured daily. The generalized linear mixed model was utilized for statistical analysis. Subdivision analysis was performed by stratifying the patients by age, sex, season, severity, the presence of family allergic diseases, sensitization, and indoor environment conditions including temperature and relative humidity. A total of 9,321 person-days of AD symptom data were collected. The average PM_2.5_ concentration was 28.7 ± 24.3 µg/m^3^, with the highest value in winter (47.1 ± 29.6 µg/m^3^). The overall effect of PM_2.5_ on AD symptoms was not statistically significant. However, an increase of 10 µg/m^3^ in indoor PM_2.5_ concentration increased AD symptom scores by 16.5% (95% CI: 6.5, 27.5) in spring and12.6% (95% CI: 4.3, 21.5) in winter, 6.7% (95% CI: 2.3, 11.3) at indoor temperatures of <25.5 °C, and by 15.0% (95% CI: 3.5, 27.7) with no use of an air purifier. The harmful effect of PM_2.5_ in boys, in children aged ≥6 years, and in children with inhalant allergen sensitization was significant, showing an increase in AD symptoms of 4.9% (95% CI: 1.4, 8.6), 12.0% (95% CI: 5.3, 19.1), and 7.0% (95% CI: 1.9, 12.3) per 10 µg/m^3^ of PM_2.5_, respectively. Furthermore, children with inhalant allergen sensitization plus severe symptoms (SCORing Atopic Dermatitis, SCORAD ≥ 30.7, median value) showed more harmful effects from exposure to PM_2.5_ (15.7% (95% CI: 4.5, 28.1) increase in AD symptom scores per 10 µg/m^3^ of PM_2.5_ increase). Indoor exposure to PM_2.5_ exacerbated AD symptoms in children in spring, winter, and at indoor temperatures of < 25.5 °C. In particular, this harmful effect was prominent in children with inhalant allergen sensitization and severe symptoms. Minimizing exposure to indoor PM_2.5_ is needed for the proper management of AD.

## 1. Introduction

Particulate matter (PM), one of the major air pollutants, is an environmental threat to human health. The biological effects of PM on health are determined largely by the size and the composition of the PM. Long-term exposure to PM with a diameter of less than 2.5 µm (PM_2.5_) has been attributed to up to 4.2 million annual premature deaths, representing 7.6% of the total global mortality [1]. PM_2.5_ deposits in the respiratory bronchioles and alveoli stimulate local and systemic inflammation as well as oxidative stress [2]. PM is associated with the increased risk of allergic diseases, such as asthma and atopic dermatitis (AD). Increased exposure to ambient PM_2.5_ significantly increased the risk of AD symptoms [3,4] and the prevalence of asthma symptoms was also associated with an increase in PM_2.5_ concentrations, even at low exposure levels under standard [5,6].

AD is a chronic inflammatory skin disease occurring mostly in early childhood and predisposes patients to food allergies and atopic march [7]. The estimated prevalence of AD in Korean children aged 18 years or younger in 2014 was about 6% [8]. One basic principle in the management of AD is avoiding aggravating factors such as microbes, allergens, emotional stress, meteorological factors, and pollutants such as PMs [3,9,10].

Most prior studies on the association between PM exposure and allergic diseases estimated the effects of ambient PM [3,4,5,11,12]. Eczema prevalence was significantly associated with long-term exposure to outdoor particulate matter with an aerodynamic diameter ≤ 10 μm (PM_10_) [11] and PM_2.5_ [12]. Several studies also found that short-term exposure to ambient PMs exacerbated the symptoms of AD patients [3,4,5,13,14]. Oh et al. (2017) [15] addressed a positive association between AD symptoms and PMs including indoor PMs. However, they measured indoor PMs at two preschools during school hours (9 a.m. to 5 p.m.) and used ambient outdoor concentrations at air quality monitoring sites during non-school hours. They did not measure residential places where children spend more time. Besides, they did not include spring and winter season when the level of PMs is the highest in Korea.

The importance of indoor air quality is, however, increasingly emphasized because people spend most of their time indoors. Korean people, for example, spend approximately 88% of their time indoors, almost 60% of which is at home on weekdays [16]. A previous study [17] demonstrated that the indoor PM components were quite different from those of outdoor PMs and the effect sizes of the indoor PM components on allergic symptoms were different from those of outdoors. However, there are very limited epidemiological studies on the short-term effects of exposure to indoor PM_2.5_ on AD symptoms, mainly because of difficulties in real-time measurements indoors.

Thus, the present study aimed to examine the short-term effect of exposure to indoor PM_2.5_ on AD symptoms in children by measuring real-time PM_2.5_ concentrations in the patients’ houses and simultaneously monitoring daily symptom scores. Subgroup analyses stratified by season, demographic status, the presence of allergen sensitization, and indoor environments were conducted to characterize the effects of indoor PM_2.5_ in children with AD.

## 2. Materials and Methods

### 2.1. Study Participants and AD Symptom Monitoring

A total of 64 children (40 boys and 24 girls) aged under 18 years with AD living in the Seoul Metropolitan Areas of Korea were enrolled in the study. They were followed up from February 2019 to November 2020. The diagnosis of AD was made according to the Hanifin and Rajka criteria [18]. AD severity was assessed using the SCORing Atopic Dermatitis (SCORAD) index [19]. Only AD patients with SCORAD scores over 15 were enrolled because patients with mild AD may not be sensitive to small changes in indoor PM_2.5_ levels. Total IgE and specific IgE against common food and inhalant allergens in peripheral blood were measured using ImmunoCAP (Thermo Fisher Scientific, Waltham, MA, USA). Sensitization was defined by specific IgE levels of over 0.69 kU/L. Common allergens included egg white, cow’s milk, soybean, wheat, peanut, *Dermatophagoides pteronyssinus*, and *D. farinae*.

The Atopic Dermatitis Symptom Score (ADSS) [20], a smartphone-based symptom diary, was used to monitor daily AD symptoms in the children. The parents or children were instructed to record AD symptoms daily using the symptom diary designed to record the extent of subjective symptoms (itching and sleep disturbance) and the degree of objective signs (erythema, dryness, oozing, and edema) on a scale of 0 to 4. The presence of AD symptoms was defined when each subjective symptom score was 2 or greater accompanied by at least two objective signs. This definition was used because our patients were not symptom-free at baseline. All patients were instructed to take a daily bath or shower and apply moisturizers frequently during the study period. The intermittent use of low potency topical corticosteroid (TCS) was allowed when needed. Along with symptom scores, the participants recorded the use of TCS and the presence of fever every day. Those who were allergic to inhalants or foods avoided exposure to the offending allergens.

At the time of participant enrollment, we also surveyed the presence of family allergic diseases (asthma, AD, and allergic rhinitis), the use of an air purifier, and demographic information.

Written informed consent was obtained from the parent or guardian of each participating child. The study protocols were reviewed and approved by the Institutional Review Board (IRB) of Samsung Medical Center (IRB No. 2018‒10-121).

### 2.2. Assessment of Indoor PM_2.5_ Exposure

To assess indoor PM_2.5_ exposure, an SSP100 (SENKO, Osan, Korea) laser-based light-scattering sensor was used. The SSP100 sensor is equipped with an IoT-based device, the Breeze^®^ (SENKO, Osan, Korea), and automatically measures PM_2.5_ every 10 min. The Breeze^®^ was placed in the living room of each participant’s house. Real-time sensor data were sent to and saved in the cloud. After retrieving the 10-min PM_2.5_ data from the cloud, we averaged the 10-min data to hourly data and compared the hourly data measured by Grimm portable aerosol spectrometer (PAS) 1.108 (Grimm Aerosol Technik GmbH & Co. KG, Ainring, Germany). We stratified the hourly data into four groups based on PM_2.5_ levels of 0–10, 10–20, 30–40, and >40 µg/m^3^ PM_2.5_. The regression parameters of the groups were obtained based on linear regression models for each group. The coefficient of determination of the regression model between the PM_2.5_ in the SSP100 and Grimm PAS was 0.93 (*p* < 0.0001) overall. We then calibrated the hourly indoor PM_2.5_ levels measured by the SSP100 in the households, applying the four parameters in accordance with the range of PM_2.5_ levels in each group. The details are shown in the Appendix A.

The average 24 h data values were calculated using the calibrated hourly PM_2.5_ and matched with the daily AD symptom data for each individual.

### 2.3. Covariates

Breeze^®^ also has temperature, relative humidity (RH), and formaldehyde sensors. Formaldehyde, indoor temperature, and indoor RH were automatically measured every 10 min. More details on indoor formaldehyde measurements are given in our previous study [21]. The average 24 h indoor temperature, indoor RH, and formaldehyde levels were calculated and used as confounding factors. For formaldehyde, the upper and lower 10% of 10 min interval data of each day were deleted and the average 24 h data value was calculated.

Ambient environment conditions such as PM_2.5_, temperature, and RH are also known to be associated with the aggravation of AD symptoms in children [22]. We obtained daily outdoor PM_2.5_ concentrations from the nearest air quality monitoring system (AQMS) station based on the home address of each participant. Daily outdoor temperature and RH were also obtained from the Korean Meteorological Administration (KMA). When the outdoor and indoor PM_2.5_, temperature and RH were matched for each individual and each day, the outdoor PM_2.5_, temperature, and RH showed strong correlations with indoor PM_2.5_, temperature, and RH. The correlation between outdoor and indoor PM_2.5_, temperature, and RH was 0.44 (*p* < 0.0001), 0.57 (*p* < 0.0001), 0.52 (*p* < 0.0001), respectively (Appendix A). Therefore, we selected indoor PM_2.5_, temperature, and RH instead of outdoor parameters to avoid the problem of collinearity between the indoor and outdoor variables, considering that the AD patients stayed indoors more than outdoors (approximately 90%) [4].

### 2.4. Statistical Analysis

Considering that repeated measurements of allergic symptoms could provide longitudinal data with a binomial distribution, the generalized linear mixed model (GLMM) with binomially distributed errors was utilized to estimate the effects of PM_2.5_ on AD symptoms. To account for serial correlation among the repeated measurements, random effects were included in the GLMM for each subject. Potential confounding factors such as age, sex, day of the week (DOW), indoor temperature and RH, indoor formaldehyde, SCORAD at enrollment, the use of TCS, and the presence of fever as a proxy of infection were controlled for when fitting the GLMM.

We also used penalized regression curves of a generalized additive mixed model (GAMM) to examine the linearity of the relationship between the exposure to indoor PM_2.5_ and AD symptoms.

For subgroup analysis, we stratified the dataset by sex, age (<6 years and ≥6 years), season, the presence of family allergic diseases, the presence of inhalant allergen sensitization, SCORAD, household income, the use of an air purifier, and indoor environmental conditions such as temperature and RH. Indoor temperature was stratified into two categories of <25.5 °C and ≥25.5 °C (median value). RH was stratified into three categories of <40%, 40–60%, and ≥60%. We then fitted GLMM models to each subgroup dataset and compared the effect sizes between them. All results are expressed as percent change with 95% confidence intervals (CIs) of symptom scores per 10 µg/m^3^ increment of PM_2.5_.

All analyses were conducted using R version 4.0.2 (The Comprehensive R Archive Network: http://cran.r-project.org, accessed on 22 June 2020). GLMMs were fitted using the lme4 package (version 3.1.2). All tests were two-sided. An alpha level of less than 0.05 was considered significant.

## 3. Results

### 3.1. Characteristics of AD Patients and Household Environments

Table 1 shows the characteristics of the subjects in the present study. The average age of the study subjects was 4.5 ± 3.7 years. The average SCORAD at enrollment was 33.2 ± 13.9. Among 64 participants, the parents or siblings of 38 (61.3%) had ever been diagnosed with allergic diseases such as asthma, allergic rhinitis, or AD. Most households had air purifiers (88.7%).

A total of 9321 person-days of symptom records were obtained from 64 subjects. Among them, 6410 (68.8%) and 2911 (30.2%) person-days were recorded by boys and girls, respectively. The presence of AD symptoms was 49.3% in 9321 person-days.

The average daily indoor temperature, RH, and formaldehyde level during the study period were 25.5 °C ± 2.6 °C, 47.2% ± 12.5%, and 13.6 ppb ± 20.1 ppb, respectively (Appendix A). With respect to the outdoor environment during the study period, the average temperature, RH, and PM_2.5_ were 14.0 °C ± 9.5 °C, 67.9% ± 15.8%, and 20.0 ± 11.8 µg/m^3^, respectively.

### 3.2. Exposure Assessment of Indoor PM_2.5_

The average indoor PM_2.5_ concentration in 9,321 datasets was 28.7 ± 24.3 µg/m^3^. Table 2 shows the indoor PM_2.5_ levels by season and indoor environment. The highest indoor PM_2.5_ level in four seasons was observed in winter (47.1 ± 29.6 µg/m^3^), followed by spring (28.0 ± 24.8 µg/m^3^), fall (22.9 ± 19.3 µg/m^3^), and summer (20.8 ± 15.0 µg/m^3^). With respect to the indoor environment, indoor PM_2.5_ concentrations were higher when the indoor temperature was below 25.5 °C (*p* < 0.0001) and the indoor RH was below 40% (*p* < 0.0001). Not surprisingly, the indoor PM_2.5_ level was lower when an air purifier was used in the home (*p* < 0.0001).

Indoor PM_2.5_ levels were compared after stratifying the patients according to their characteristics (Table 3). The indoor PM_2.5_ levels were higher in boys, children with family allergic diseases, and children with inhalant allergen sensitization than in girls (*p* < 0.0001), children without a family history (*p* < 0.0001), and children without allergen sensitizations (*p* = 0.0026), respectively. In children with SCORAD ≥ 40.9, indoor PM_2.5_ levels were lower than in children with mild AD (*p* = 0.0003), suggesting that the more severe the AD of their child, the better indoor environment the parents were trying to make.

### 3.3. Effects of PM_2.5_ on AD Symptoms

Figure 1 shows the relationship between AD symptoms and indoor PM_2.5_ levels as a result of GAMM-fitting after controlling for confounding factors including indoor formaldehyde, temperature, RH, SCORAD at enrollment, fever, DOW, and the use of TCS. The spline curves show that the AD symptoms increased monotonically in spring and winter as the level of indoor PM_2.5_ increased, whereas there was no linear relationship in summer and the relationship was negative in fall.

Figure 2 and Figure 3 show the effects of indoor PM_2.5_ exposure on AD symptoms as a result of GLMM-fitting after controlling for confounding factors including indoor formaldehyde, temperature, RH, SCORAD at enrollment, fever, DOW, and the use of TCS. An increase in PM_2.5_ of 10 µg/m^3^ was associated with a 0.9% (95% CI: −2.1, 4.0) increase in AD symptoms as a whole. The overall effect of PM_2.5_ on AD symptoms was not statistically significant. However, when the patients were stratified by season, the PM_2.5_ effects on AD symptoms were statistically significant in spring and winter, showing a higher effect size in spring (Figure 2). With an increase in PM_2.5_ of 10 µg/m^3^, the presence of AD symptoms increased by 16.5% (95% CI: 6.5, 27.5) in spring and 12.6% (95% CI: 4.3, 21.5) in winter when indoor formaldehyde, temperature, and RH were controlled for. Low indoor temperature was also a risk factor to exacerbate the adverse effects of PM_2.5_ on AD symptoms in children (Figure 2). When the indoor temperature was below 25.5 °C, an increase in PM_2.5_ of 10 µg/m^3^ increased AD symptom scores by 6.7% (95% CI: 2.3, 11.3). AD symptom scores in children living in houses without an air purifier were significantly increased by 14.9% (95% CI: 3.5, 27.7) by an increase in indoor PM_2.5_ of 10 µg/m^3^, whereas there was no increase in children living in houses with an air purifier. All numbers for the effect sizes are shown in Appendix A.

The harmful effect of PM_2.5_ in boys was significant, showing an increase in AD symptom scores of 4.9% (95% CI: 1.4, 8.6) per 10 µg/m^3^ of PM_2.5_. AD symptom scores in children aged 6–18 years increased significantly by 12.0% (95% CI: 5.3, 19.1) by an increase in indoor PM_2.5_ levels of 10 µg/m^3^, whereas there was no increase in young children under 6 years. AD symptom scores in children with inhalant allergen sensitization increased by 7.0% (95% CI: 1.9, 12.3) per 10 µg/m^3^ of PM_2.5_ increase. Notably, the harmful effect of PM_2.5_ on AD symptoms was greater in children with inhalant allergen sensitization plus severe symptoms (SCORAD at enrollment ≥ 30.7), showing 15.7% (95% CI: 4.5, 28.1) per 10 µg/m^3^ of PM_2.5_ (Figure 3).

## 4. Discussion

Although air pollution is considered to be a significant risk factor for AD [9], there is a paucity of studies investigating the effect of indoor PM on AD symptoms. In the present study, we longitudinally followed AD patients to assess the effects of indoor PM_2.5_ exposure on AD symptoms in children based on repeated measurements of indoor PM_2.5_ using a real-time sensor device. To overcome the issue of measurement data accuracy, we calibrated and validated the measured data based on standard measurement devices (details are in Appendix A). Furthermore, the effect of indoor PM_2.5_ was adjusted by indoor temperature, RH, and formaldehyde, an important indoor air pollutant, in our statistical models. Finally, we obtained results on the environmental factors affecting indoor PM_2.5_ levels as well as the characteristics of the patients sensitively responding to changes in indoor PM_2.5_ concentrations. To the best of our knowledge, this was the first study on the association between indoor PM_2.5_ and AD symptoms in children based on real-time indoor PM_2.5_ measurements in residential places. Our results can be used in real clinical practice for the management of moderate-to-severe AD in children.

Indoor PM originates partly from the outside, depending upon the frequency of ventilation, whereas there are many indoor sources of PM as well [9]. The reason why the indoor PM_2.5_ levels were higher in winter and spring than in summer and fall in the present study can be related to less frequent ventilation. In summer, 39 of 62 participants (63%) ventilated their houses by opening windows every day, compared to only 24 (39%) in winter. Consequently, the correlation coefficient of indoor and outdoor PM_2.5_ concentrations in summer was 0.47 (*p* < 0.0001), whereas they were 0.32 and 0.22 in winter and spring, respectively (Appendix A). These findings imply that the higher the number of natural ventilation events, the lower the indoor PM_2.5_ levels. According to previous studies [23,24], indoor air quality in homes in urban areas was influenced by indoor human activities such as cooking, smoking, and cleaning. For example, high peak PM episodes were observed during cooking activities in Seoul [23]. Natural ventilation may be necessary to lower the levels of PM indoors even in winter and spring. In this study, indoor PM_2.5_ levels were higher when the indoor temperature and RH were lower. This can be related to relatively lower indoor temperatures and RH in winter (23.5 ± 2.1 °C, 37.6 ± 8.8%) than in summer (27.3 ± 20.0 °C, 58.6 ± 8.9%) (*p* < 0.0001). Thus, when indoor PM_2.5_ concentrations increase by the influence of the season, indoor temperature, or RH, it is highly likely that AD symptoms worsen according to the exposure concentration.

In the present study, we observed that AD symptom scores were significantly increased by an increase in indoor PM_2.5_ levels in specific seasons such as spring and winter, at indoor temperatures of <25.5 °C, and in houses without air purifiers. Perhaps the reason is that indoor PM_2.5_ concentrations were higher under these conditions (Table 2) and implies that it is necessary to monitor whether indoor PM_2.5_ concentrations increase in such environmental conditions. We also found that the AD symptom scores increased, particularly in boys compared to girls, and the PM_2.5_ levels in households where boys lived were higher (30.7 ± 25.2 µg/m^3^) than in those where girls lived (24.4 ± 21.6 µg/m^3^, *p* < 0.0001). These findings explain why boys with AD were more vulnerable to PM_2.5_ exposure than girls in the present study. The reason why the association between PM_2.5_ and AD flares in girls was negative is not clear. Behavioral differences might be one of the reasons. Parents having girls with severe AD may have a tendency to manage AD symptoms more strictly when the PM2.5 level is high. However, considering that there are biological differences between boys and girls in immune development, infection, and gut microbiota [25,26,27], it is not clear whether our results were simply due to differences in PM_2.5_ concentrations. Further investigations are needed to explain the sex-specific response to indoor PM_2.5_.

It is necessary to identify the environmental factors affecting indoor PM_2.5_ levels, but it is also important to identify which patients with AD in clinical practice are more susceptible to PM_2.5_ exposure, even at low concentrations. Indeed, our previous study demonstrated that responses to outdoor PM_10_ exposure varied considerably among individuals with AD [28]. In the present study, it was notable that the harmful effect of PM_2.5_ in children with inhalant allergen sensitization was significant, with increases in AD symptoms of 7.0% (95% CI: 1.9, 12.3) per 10 µg/m^3^ of PM_2.5_. This could be because house dust mite allergens are one of the components of indoor fine dust, unlike outdoor PM. Another single risk factor, severe AD (SCORAD ≥ 30.7, median value), did not show statistical significance in AD symptom aggravation. However, when combined with house dust mite sensitization, AD symptom scores significantly increased by 15.7% (95% CI: 4.5, 28.1) per 10 µg/m^3^ of PM_2.5_. This finding indicates that AD severity could be a risk factor in children who are sensitized to house dust mite allergen. Therefore, these host factors including house dust mite sensitization and disease severity should be considered as risk factors affecting host susceptibility to PM_2.5_ exposure in children with AD.

We observed that children aged over 6 years showed a significant increase in percent change in AD symptoms by PM_2.5_ exposure, whereas those under 6 years did not. As shown in Appendix A, there was no difference in PM_2.5_ levels between the two groups (*p* = 0.4362). When we compared the host factors between children aged ≥6 years and <6 years, there seemed to be a difference in AD severity, inhalant allergen sensitization, and body mass index (BMI), but was no statistical significance. It is speculated that different behaviors according to age may explain this difference. Children over age 6 years may participate in more vigorous activities indoors and spend more time outdoors where they are exposed to a variety of environmental triggers compared to children under age 6 years. Unfortunately, we did not find the exact reason to explain the age-related difference and further studies with more patients are needed.

A systematic review and meta-analysis demonstrated that PM_2.5_ was associated with the aggravation of AD symptoms at a young age (OR = 1.05; 95% CI: 0.95–1.16; I^2^ = 46%) [29]. The exact mechanisms through which PM can aggravate AD symptoms are not fully understood. It has been reported that the skin barrier is damaged by direct exposure to PM. A high PM_2.5_ level might generate reactive oxygen species (ROS) and induce oxidative damage to proteins in keratinocytes and eventually, lead to the disruption of the skin barrier and further exacerbation of AD [30,31]. Jin et al. [32] showed that cellular ROS production was increased by PM treatment in mice, and antioxidant N-acetyl cysteine pretreatment prevented the induction of IL-8 and matrix metalloprotease 1. A recent study demonstrated that PM_2.5_-induced tumor necrosis factor-α downregulated filaggrin expression in human keratinocytes through the aryl hydrocarbon receptor (AhR) pathway and subsequently, caused skin barrier dysfunction [33]. It is suggested that polycyclic hydrocarbons (PAHs), a component of airborne fine particles, contributed to the activation of the AhR signaling pathway and the transactivation of neurotrophic factor artemin [34,35]. The intercellular penetration of PM was seen in barrier-disrupted skin as well [32].

One of the limitations of this study was that we did not include outdoor PM_2.5_ levels in the exposure assessments. Considering that the AD patients spend over 90% of their time indoors [4] and that the AQMS-monitored PM_2.5_ could have led to misclassification bias of the participants whose residences were far from the monitoring station [36], we included only indoor PM_2.5_ measured in the households, with no consideration of outdoor exposure. With more spatially resolved outdoor monitoring data, PM_2.5_ exposure assessment could be improved. Another limitation is that we did not consider the effect of environmental tobacco smoke (ETS) exposure as a covariate. However, we assumed that our patients were not exposed to ETS because the parents classified as current smokers reported that they did not smoke in the home. We had many missing data in daily symptom reports during the study period; the average count of daily reports was 153 ± 89 days, although children or parents were guided to report every day for a continual year. This might cause the bias of PM effect on AD symptoms. Finally, caution is needed in generalizing our results because the number of patients was small and our study was limited to a specific race, age, region, and time period.

## 5. Conclusions

In conclusion, indoor PM_2.5_ exposure exacerbated AD symptoms in children in spring, winter, and at indoor temperatures of <25.5 °C. For the proper management of AD, indoor PM_2.5_ must be considered as one of the important environmental triggers, and minimizing the exposure to indoor PM2.5 should be individualized because host characteristics and surrounding environmental factors differ from patient to patient. Ventilation, proper use of air purifiers, and maintenance of proper indoor temperature and RH can be helpful to minimize the indoor PM_2.5_ effect on AD symptoms. In particular, this harmful effect was prominent in children with inhalant allergen sensitization and severe symptoms. The control of indoor PM_2.5_ should be considered in the management of AD symptoms particularly in children with inhalant allergen sensitization or with severe symptoms.

## Figures and Tables

**Figure 1 ijerph-18-11509-f001:**
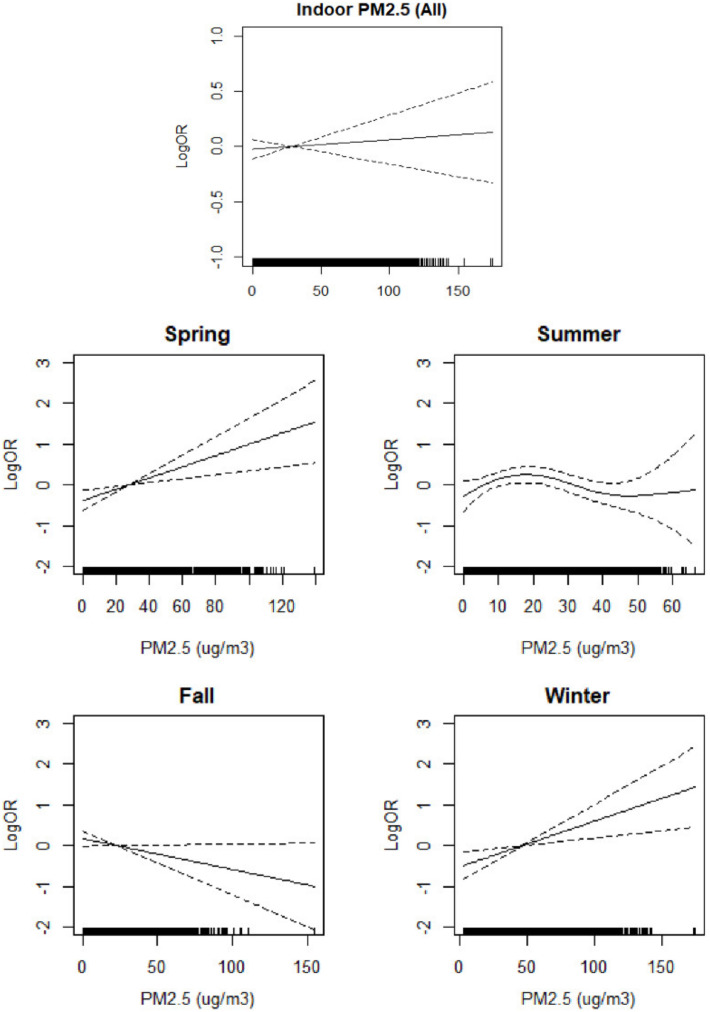
Relationship of indoor PM_2.5_ with atopic dermatitis (AD) symptoms. Each figure shows the spline curves (solid lines) with 95% confidence intervals (two dashed lines). The model was controlled for the severity score at the initial visit, age, sex, presence of fever, and day of week (DOW), indoor RH, temperature, and formaldehyde; OR: odds ratio.

**Figure 2 ijerph-18-11509-f002:**
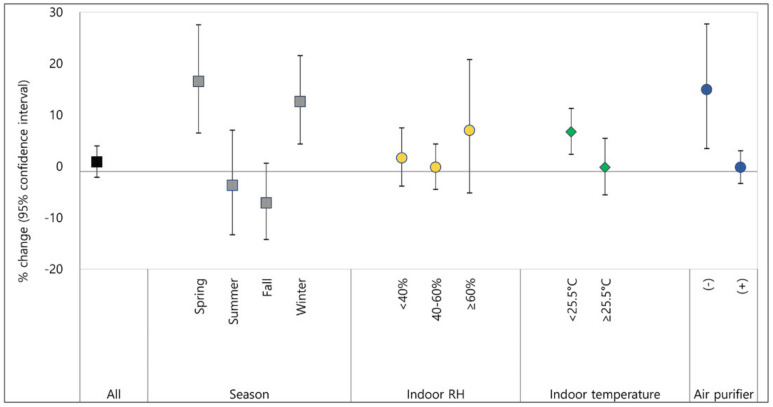
Percent changes of atopic dermatitis symptoms caused by PM_2.5_ exposure. % change in AD symptoms per 10 µg/m^3^ of PM_2.5_ exposure; dot or square indicate % change in AD symptoms per 10 µg/m^3^ of PM_2.5_ exposure and lines are 95% confidence interval; RH, relative humidity.

**Figure 3 ijerph-18-11509-f003:**
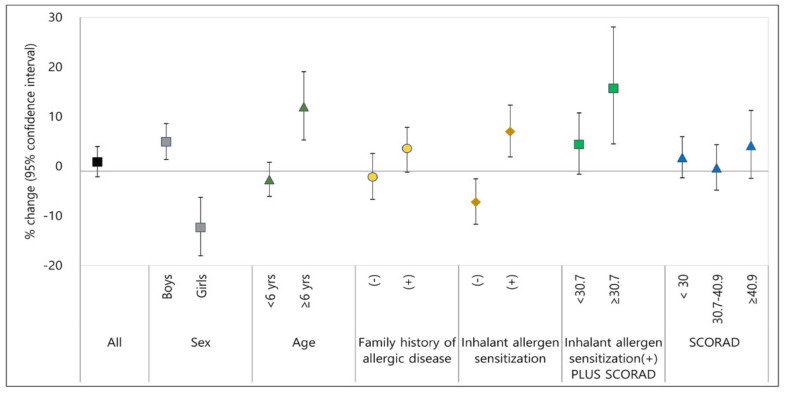
Percent changes of atopic dermatitis symptoms caused by PM_2.5_ exposure. % change in AD symptoms per 10 µg/m^3^ of PM_2.5_ exposure; dot or square indicate % change in AD symptoms per 10 µg/m^3^ of PM_2.5_ exposure and lines are 95% confidence interval; SCORAD, SCORAD at enrollment.

**Table 1 ijerph-18-11509-t001:** Characteristics of the subjects in this study (*n* = 64) *.

Characteristics	Total
Sex (boys:girls)	40:24
Age (yrs)	4.5 ± 3.7
BMI (kg/m^2^)	17.1 ± 2.4
SCORAD at enrollment	33.2 ± 13.9
Total IgE (U/L)	799.1 ± 1392.5
Sensitization, n (%)	50/64 (78.1%)
Food allergens ^‡^	47/64 (71.2%)
Inhalant allergens ^§^	23/64 (46.0%)
Presence of family allergic diseases, n (%)	38/62 (61.3%)
Use of air purifier, n (%)	55/62 (88.7%)
No. of record (person-days)	9321
Presence of AD symptoms (%)	49.3
Presence of fever (%)	0.9
Use of TCS (%)	53.3

* Data are expressed as the mean ± standard deviation; ^‡^ Sensitized by five common food allergens, including egg white, cow’s milk, soybean, wheat, or peanut; ^§^ Sensitized by house dust mite (*Dermatophagoides pteronyssinus*, *D. farinae*); BMI, body mass index; SCORAD, SCORing of Atopic Dermatitis; TCS, topical corticosteroid.

**Table 2 ijerph-18-11509-t002:** Summary of indoor PM_2.5_ concentration by season and indoor environments (*n* = 64).

Classification	Subgroups	Number (Person-Days)	Mean ± SD (µg/m^3^)	*p*-Value *
All		9321	28.7 ± 24.3	
Season	Spring	2093	28.0 ± 24.8	-
	Summer	2443	20.8 ± 15.0	<0.0001
	Fall	2781	22.9 ± 19.3	<0.0001
	Winter	2004	47.1 ± 29.6	<0.0001
Indoor RH	<40%	2748	36.5 ± 28.1	-
	40–60%	5057	26.0 ± 23.3	<0.0001
	≥60%	1516	25.5 ± 21.7	<0.0001
Indoor temperature	<25.5 °C	4570	35.4 ± 27.4	<0.0001
	≥25.5 °C	4751	22.3 ± 18.7	
Air purifier	(−)	785	39.9 ± 28.2	<0.0001
	(+)	8429	27.8 ± 23.7	

* ANOVA test for season and RH and z-test for others; RH, relative humidity.

**Table 3 ijerph-18-11509-t003:** Summary of indoor PM_2.5_ concentration by subgroup (*n* = 64).

Classification	Subgroups	Number (Person-Days)	Mean ± SD (µg/m^3^)	*p*-Value *
All		9321	28.7 ± 24.3	
Sex	Boys	6410	30.7 ± 25.2	<0.0001
	Girls	2911	24.4 ± 21.6	
Age	<6 yrs	6924	28.6 ± 24.4	0.4362
	≥6 yrs	2397	29.1 ± 23.9	
Family history of allergic diseases	(−)	3474	25.8 ± 22.2	<0.0001
	(+)	5740	30.7 ± 25.4	
Inhalant allergen sensitization	(−)	3501	28.2 ± 25.4	0.0026
	(+)	3908	30.0 ± 23.7	
SCORAD at enrollment	<30.7	4624	28.4 ± 24.7	-
	30.7–40.9	2269	29.6 ± 24.8	0.0525
	≥40.9	2428	26.7 ± 21.4	0.0003
Inhalant allergen sensitization (+) PLUS SCORAD	<30.7 (median)	2379	31.2 ± 24.5	<0.0001
	≥30.7 (median)	1529	28.0 ± 22.3	

* Z-test for mean differences between subgroups except for SCORAD at enrollment which was result from ANOVA test. SCORAD, SCORing of Atopic Dermatitis.

## Data Availability

Data can be accessed upon request via correspondent.

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
