# Peer review of "Effects of Exposure to Indoor Fine Particulate Matter on Atopic Dermatitis in Children"

_ijerph, 2021, doi:10.3390/ijerph182111509_

Round 1

Reviewer 1 Report

The manuscript  „EFFECTS OF EXPOSURE TO INDOOR FINE PARTICULATE MATTER ON ATOPIC DERMATITIS IN CHILDREN“
presents the analysis of specific parameters  (the short-term effect of exposure to indoor fine particulate matter)  in children with atopic dermatitis
and gives scientifically very useful data showing significant negative effect of particulate matter exposure on atopic dermatitis symptoms.
As a clinician, I consider this research and the manuscript well done and original, giving unique insight on the effect of indoor particulate matter
on atopic dermatitis. Excluding the effect of outdoor particulate matter exposure is the biggest limitation of the study. Although authors claim that
participants spent more than 90% of their time indoors, still the effect of outdoor particulate matter exposure was, to date, proven to be significant
on atopic dermatitis.  Author stated this limitation properly in the discussion of the manuscript.  As the minor recommendation, the authors may add more data considering the use of their results in the practice and their meaning for the further patient management.

Author Response

The manuscript  “EFFECTS OF EXPOSURE TO INDOOR FINE PARTICULATE MATTER ON ATOPIC DERMATITIS IN CHILDREN” presents the analysis of specific parameters  (the short-term effect of exposure to indoor fine particulate matter)  in children with atopic dermatitis and gives scientifically very useful data showing significant negative effect of particulate matter exposure on atopic dermatitis symptoms.
As a clinician, I consider this research and the manuscript well done and original, giving unique insight on the effect of indoor particulate matter on atopic dermatitis. Excluding the effect of outdoor particulate matter exposure is the biggest limitation of the study. Although authors claim that participants spent more than 90% of their time indoors, still the effect of outdoor particulate matter exposure was, to date, proven to be significant on atopic dermatitis. Author stated this limitation properly in the discussion of the manuscript.  As the minor recommendation, the authors may add more data considering the use of their results in the practice and their meaning for the further patient management.

Answer: Thank you for the comments. We have added the meaning of the results of the current study in conclusion. [Line 364-367]

Reviewer 2 Report

This panel study investigated the short-term effect of exposure to indoor PM2.5 on AD symptoms in 64 children with AD. Overall, the manuscript is well written and of good quality. There are several minor comments to consider when revising the manuscript. 

Introduction

Lines 51-52: "Most prior studies on the association between PM exposure and allergic diseases estimated the effects of ambient PM [3-5]."

Two of the three cited articles were from the same research group. More balanced choices are needed. Please refer to the following articles.

Penard-Morand et al., 2010, Long-term exposure to close-proximity air pollution and asthma and allergies in urban children Song et al., 2011, Acute health effects of urban fine and ultrafine particles on children with atopic dermatitis
Kathuria and Silverberg, 2016, Association of pollution and climate with atopic eczema in US children Baek et al., 2021, Associations between ambient air pollution and medical care visits for atopic dermatitis   Please explain what is the difference between the current study and a similar previous study entitled "Association between particulate matter concentration and symptoms of atopic dermatitis in children living in an industrial urban area of South Korea" by Oh et al. (2018). This previous study also measured indoor PM2.5.     Methods  The authors stated that the study participants were followed up from February 2019 to November 2020. My understanding was that the 64 children submitted daily AD symptom reports during this period of 22 months. However, the person-days was only 9,321. I suppose there were many missing days during the study period.    Discussion It would be worth discussing the inverse association in girls, which seems to be reverse causation. One of the explanations is that their parents had a tendency to manage AD symptoms more strictly when the PM2.5 level is projected to be high. The above mentioned issue of missing reports is one of the explanations as well, particularly when the pattern of missing is related to symptom occurrence.

Author Response

This panel study investigated the short-term effect of exposure to indoor PM2.5 on AD symptoms in 64 children with AD. Overall, the manuscript is well written and of good quality. There are several minor comments to consider when revising the manuscript. 

 Introduction

Lines 51-52: "Most prior studies on the association between PM exposure and allergic diseases estimated the effects of ambient PM [3-5]."

Two of the three cited articles were from the same research group. More balanced choices are needed. Please refer to the following articles. Penard-Morand et al., 2010, Long-term exposure to close-proximity air pollution and asthma and allergies in urban children; Song et al., 2011, Acute health effects of urban fine and ultrafine particles on children with atopic dermatitis; Kathuria and Silverberg, 2016, Association of pollution and climate with atopic eczema in US children.

Answer: Thank you for your kind recommendation. We have added the articles in Introduction. [Line 52-55]

Please explain what is the difference between the current study and a similar previous study entitled "Association between particulate matter concentration and symptoms of atopic dermatitis in children living in an industrial urban area of South Korea" by Oh et al. (2018). This previous study also measured indoor PM2.5.    

Answer: They measured indoor PMs at two classrooms during school hours (9 a.m. to 5 p.m.) and used ambient outdoor concentrations at two air quality monitoring sites during non-school hours. They did not measure PMs in residential places where children spend more time. We have cited the article in the manuscript and mentioned a difference between the article and our current study. [Line 56-61]

Methods 

The authors stated that the study participants were followed up from February 2019 to November 2020. My understanding was that the 64 children submitted daily AD symptom reports during this period of 22 months. However, the person-days was only 9,321. I suppose there were many missing days during the study period.

Answer: Although we tried to collect daily symptom scores for one year per person, our final average of reported days was 153±89 days (Range: 29~ 395 days). As the reviewer indicated, we had many missing data during the study period. We have added this limitation in Discussion. [Line 356-359]  

Discussion

It would be worth discussing the inverse association in girls, which seems to be reverse causation. One of the explanations is that their parents had a tendency to manage AD symptoms more strictly when the PM2.5 level is projected to be high. The mentioned issue of missing reports is one of the explanations as well, particularly when the pattern of missing is related to symptom occurrence.

Answer: We agree with the reviewer’s opinion. In the Discussion, we have mentioned what the reviewer indicated. [Line 300-302]

Reviewer 3 Report

This is a prospective cohort study evaluating the short-term effect of exposure to indoor PM2.5 on atopic dermatitis (AD) symptoms among 64 children with moderate-to-severe AD. The study followed up the children from Feb 2019 to Nov 2020. Indoor PM2.5 concentration was monitored using a sensor which measured PM2.5 every 10-min (and the concentration was later extrapolated to daily average concentration for analysis). The outcome, SCORAD (atopic dermatitis symptom score), was measured at baseline as well we daily during the follow up using a smartphone-based diary. Baseline characteristics including age, gender, family history of allergic diseases, inhalant allergen sensitization, presence of fever, and TCS ( use. Using a GLMM (generalized linear mixed model), the authors found a positive association between PM2.5 and SCORAD in certain subgroups but not others.

  1. My main comment is the heterogeneity or inconsistency of findings between groups casts doubt on the association between PM2.5 and AD symptom worsening. It seems positive association was only found among those with higher PM2.5, though the current study could not test such hypothesis due to the limited sample size. Given such, the conclusions on the association between PM2.5 and AD conditional on other factors might be premature and due to chance. This should be at least declared as a limitation.
  2. Do you have information or literature data on how fast (within a day or with delay) the body responds to increase in PM2.5 in terms of AD symptoms from clinical perspective?

Author Response

This is a prospective cohort study evaluating the short-term effect of exposure to indoor PM2.5 on atopic dermatitis (AD) symptoms among 64 children with moderate-to-severe AD. The study followed up the children from Feb 2019 to Nov 2020. Indoor PM2.5 concentration was monitored using a sensor which measured PM2.5 every 10-min (and the concentration was later extrapolated to daily average concentration for analysis). The outcome, SCORAD (atopic dermatitis symptom score), was measured at baseline as well we daily during the follow up using a smartphone-based diary. Baseline characteristics including age, gender, family history of allergic diseases, inhalant allergen sensitization, presence of fever, and TCS use. Using a GLMM (generalized linear mixed model), the authors found a positive association between PM2.5 and SCORAD in certain subgroups but not others.

  1. My main comment is the heterogeneity or inconsistency of findings between groups casts doubt on the association between PM2.5 and AD symptom worsening. It seems positive association was only found among those with higher PM2.5, though the current study could not test such hypothesis due to the limited sample size. Given such, the conclusions on the association between PM2.5 and AD conditional on other factors might be premature and due to chance. This should be at least declared as a limitation.

Answer: Thank you for the comment. We have added the limitation in the Discussion. (Line 359-361)

  1. Do you have information or literature data on how fast (within a day or with delay) the body responds to increase in PM2.5 in terms of AD symptoms from clinical perspective?

Answer: The effect of PMs on AD symptom tends to be acute within a day and lasting 3 days. In our previous study1, we investigated the lagged effect of PMs on AD symptom using daily symptom data. We found that the adverse effect of ambient PM10 appeared on same day and also with delays up to 3 days. Please refer to our previous study below.

  1. Kim, Y.M.; Kim, J.; Han, Y.; Lee, B.J.; Choi, D.C.; Cheong, H.K.; Jeon, B.H.; Oh, I.; Bae, G.N.; Lee, J.Y.; et al. Comparison of diverse estimation methods for personal exposure to air pollutants and associations with allergic symptoms: The Allergy & Gene-Environment Link (ANGEL) study. Total Environ. 2017, 579, 1127-1136.
